# Structural Moieties Required for Cinnamaldehyde-Related Compounds to Inhibit Canonical IL-1β Secretion

**DOI:** 10.3390/molecules23123241

**Published:** 2018-12-07

**Authors:** Su-Chen Ho, Yi-Huang Chang, Ku-Shang Chang

**Affiliations:** Department of Food Science, Yuanpei University of Medical Technology, No. 306, Yuanpei Street, Hsinchu 300, Taiwan; yihuang@mail.ypu.edu.tw (Y.-H.C.); kushang@mail.ypu.edu.tw (K.-S.C.)

**Keywords:** cinnamaldehyde, 2-methoxy cinnamaldehyde, NLRP3 inflammasome, IL-1β, sterile inflammation

## Abstract

Suppressing canonical NOD-like receptor protein 3 (NLRP3) inflammasome-mediated interleukin (IL)-1β secretion is a reliable strategy for the development of nutraceutical to prevent chronic inflammatory diseases. This study aimed to find out the functional group responsible for the inhibitory effects of cinnamaldehyde-related compounds on the canonical IL-1β secretion. To address this, the suppressing capacities of six cinnamaldehyde-related compounds were evaluated and compared by using the lipopolysaccharide (LPS)-primed and adenosine 5′-triphosphate (ATP)-activated macrophages. At concentrations of 25~100 μM, cinnamaldehyde and 2-methoxy cinnamaldehyde dose-dependently inhibited IL-1β secretion. In contrast, cinnamic acid, cinnamyl acetate, cinnamyl alcohol and α-methyl cinnamaldehyde did not exert any inhibition. Furthermore, cinnamaldehyde and 2-methoxy cinnamaldehyde diminished expressions of NLRP3 and pro-IL-1β. Meanwhile, cinnamaldehyde and 2-methoxy cinnamaldehyde prevented the ATP-induced reduction of cytosolic pro-caspase-1 and increase of secreted caspase-1. In conclusion, for cinnamaldehyde-related compounds to suppress NLRP3 inflammasome-mediated IL-1β secretion, the propenal group of the side chain was essential, while the substituted group of the aromatic ring played a modifying role. Cinnamaldehyde and 2-methoxy cinnamaldehyde exerted dual abilities to inhibit canonical IL-1β secretion at both stages of priming and activation. Therefore, there might be potential to serve as complementary supplements for the prevention of chronic inflammatory diseases.

## 1. Introduction

Inflammation is an indispensable immunological response of host to against pathogenic organism and injured challenge. However, inappropriate inflammation generates excess amounts of inflammatory cytokines which will cause tissue destruction and ultimately result in severe disorders. Interleukin (IL)-1β is a chief orchestrating inflammatory cytokine, exerting a broad range of immune actions, such as activation of cells to produce other inflammatory cytokines and chemokines, induction of endothelial cells to express cell membrane adhesion molecules to recruit leucocytes and cooperation of performances and differentiation of inherent and adaptive lymphoid cells [1,2,3,4]. Because of these multiple widely influencing inflammatory functions, the canonical secretion of IL-1β is tightly regulated by a distinct two-signal procedure. Additionally, inflammasome, a multi-protein complex, which is composed of cytosolic pattern recognition receptors (PRRs), adaptor proteins- apoptosis-associated speck-like containing a C-terminal caspase recruitment domain (CARD) (ASC) and pro-caspase-1, is involved in the regulation. The first stage of canonical IL-1β secretion is referred to as priming. Through engagement with PRRs on the membrane such as toll like receptor-4 (TLR-4), the first signal of pathogen-associated molecular patterns (PAMPs) is recognized, thereby, macrophages are primed to elicit the intracellular NF-κB signaling pathway which further drives the synthesis of immature precursor of IL-1β (pro-IL-1β) and inflammasome components. The second stage is referred to as activation. When the second signal of damage associated molecular patterns (DAMPs) or PAMPs is sensed by cytosolic PRRs, it leads to the assembly of the inflammasome complex. After that, caspase-1 is activated by auto-cleavage. The pro-IL-1β is then hydrolyzed by caspase-1 into their mature forms that are rapidly secreted [5].

The inflammasome constructed by the NOD–like receptors (NLRs) family, especially NOD-like receptor protein 3 (NLRP3), is the most fully characterized inflammasome. The NLRP3 can sense diverse endogenous DAMPs, including extracellular adenosine 5′-triphosphate (ATP), β-amyloid plaque, uric acid crystal and cholesterol crystal, to be activated. Additionally, the activated NLRP3 inflammasome executes maturation of IL-1β and results in a so-called “sterile inflammation”. Such endogenous DAMP-involved inflammation has been demonstrated to be closely related to several non-communicable diseases, such as type 2 diabetes, Alzheimer’s disease, arthritis, gout and atherosclerosis. Consequently, the inhibition of NLRP3 mediated IL-1β secretion is now generally considered as an effective strategy to prevent sterile inflammation associated diseases [6,7,8].

On the other hand, cinnamaldehyde inhibits generation of pro-inflammatory cytokines and factors such as IL-1β, TNF-α and nitric oxide, etc., through suppressing NF-κB signaling, and is a putative anti-inflammatory phytochemical [9]. Cinnamaldehyde has been demonstrated to suppress the endotoxin-induced TLR-4 activation in macrophages [10]. Moreover, Cinnamaldehyde also inhibits upstream NF-κB-inducing kinase (NIK)/IκB kinase (IKK), extracellular signal-related kinase (ERK) and p38 mitogen-activated protein kinase (MAPK) of the NF-κB signaling pathway in aged rats [11]. The relationship between the inhibitory activity on the NF-κB governed the inflammatory response and the structure of cinnamaldehyde-related compounds is well characterized. Some synthetic derivatives have been developed with more activity and less toxicity [12]. However, thus far, it is unclear which functional group of the cinnamaldehyde-related compounds are crucially responsible for the inhibition of NLRP3 inflammasome-mediated canonical IL-1β secretion. To address this, the suppressing capacity of six cinnamaldehyde-related compounds on canonical IL-1β secretion were evaluated by using a lipopolysaccharide (LPS)-primed and ATP-induced macrophage model. Simultaneously, the underlying molecular mechanism was investigated.

## 2. Results

### 2.1. Inhibitory Capacities of the Cinnamaldehyde-Related Compounds on Canonical IL-1β Secretion

In this study, six cinnamaldehyde-related compounds were selected and their structures are shown in Figure 1a. In order to prevent the overdose-induced cytotoxicity, the MTT assay was first carried out to assess the reasonable treatment concentrations of the cinnamaldehyde-related compounds. As shown in Figure 1b, at a concentration of 200 μM, cinnamaldehyde and 2-methoxy cinnamaldehyde caused a lower cellular survival (<95%). However, at concentrations of 100 μM and bellow, all the tested compounds did not lead to cytotoxicity. Consequently, the capacities of the cinnamaldehyde-related compounds to inhibit IL-1β secretion were assessed at concentrations ≤100 μM. Figure 2a presents the effect of the cinnamaldehyde-related compounds on the IL-1β secretion in LPS-primed and ATP-activated macrophages. After LPS-priming and ATP-activation, IL-1β secretion dramatically elevated from 143 to 8124 pg/mL. Among the tested compounds, only cinnamaldehyde and 2-methoxy cinnamaldehyde suppressed IL-1β secretion in a dose-dependent manner at concentrations of 25~100 μM. At 100 μM, cinnamaldehyde and 2-methoxy cinnamaldehyde completely blocked IL-1β secretion. Additionally, cinnamaldehyde and 2-methoxy cinnamaldehyde diminished the secretion of pro-inflammatory TNF-α, of which the expression is primarily governed by NF-κB signaling (Figure 2b). At 100 μM, α-methyl cinnamaldehyde mildly attenuated TNF-α, but not IL-1β secretion. The other compounds, including cinnamic acid, cinnamyl alcohol, and cinnamyl acetate, diminished neither IL-1β secretion nor TNF-α secretion. The results indicated that the aldehyde group of the side chain is the most critical determinant for inhibition of cinnamaldehyde-related compounds on the canonical IL-1β secretion. Alkyl substitution on the α-carbon of the side chain severely impacted the inhibitory capacity of cinnamaldehyde-related compounds, while methoxy substitution on the 2-carbon of benzyl group seemed to slightly improve.

In order to explore the inhibitory effect of these compounds on the maturation stage of canonical pathway of IL-1β, macrophages were primed before compound treatment and ATP activation. As shown in Figure 3, without ATP activation, LPS-primed cells could secrete a moderate amount (1415 pg/mL) of IL-1β. In fact, in addition to the canonical pathway, a non-canonical IL-1β secretion has been demonstrated [13]. The TRLs (Toll-like receptors) ligands, such as LPS, can trigger activation of Fas signaling, which will further activate caspase-8 through the adapter molecule Fas-associated death domain (FADD). The active capase-8 then cleavage immature pro-IL-1β into mature IL-1β. Consequently, as shown here, a single LPS-priming was enough to make macrophages to secret IL-1β through non-canonical pathway. Despite of this, ATP activation further enhanced IL-1β secretion to 5745 pg/mL and this increased IL-β should originated from the canonical NLRP3 inflammasome pathway. Among the tested compounds, only cinnamaldehyde and 2-methoxy cinnamaldehyde significantly attenuated ATP-stimulated IL-1β secretion. The results implied that cinnamaldehyde and 2-methoxy cinnamaldehyde could inhibit the partially maturation process of the canonical pathway.

### 2.2. Influence of Cinnamaldehyde and 2-Methoxy Cinnamaldehyde on the LPS-Primed mRNA Expression of IL-1β and NLRP3 Inflammasome-Related Components

In the following experiments, the two potent cinnamon phytochemicals, cinnamaldehyde and 2-methoxy cinnamaldehyde, and one of the ineffective cinnamon phytochemicals as a negative reference, namely, α-methyl cinnamaldehyde, were selected to simultaneously compare and investigate the molecular mechanism. To confirm the suppressing effect of cinnamaldehyde and 2-methoxy cinnamaldehyde on the priming stage, the mRNA levels of IL-1β, NLRP3 and other inflammasome-related components were determined. Figure 4 presents the influence of the chosen compounds on the LPS-primed mRNA expression of TNF-α, IL-1β and NLRP3 inflammsome related genes, such as caspase-1, ASC and ATP-gated purinergic P2X7 receptor (P2X7R). Undoubtedly, LPS priming evidently induced TNF-α, IL-1β, and NLRP3 mRNA expressions. In contrast, caspase-1, ASC and P2X7R expressions were not affected by LPS-priming. Both of cinnamaldehyde and 2-methoxy cinnamaldehyde reduced the LPS-triggered TNF-α, IL-1β and NLRP3 mRNA expressions. Unsurprisingly, α-methyl cinnamaldehyde did not influence any of the LPS-primed gene expressions. The results indicated that suppressing LPS-primed IL-1β and NLRP3 mRNA expressions contributed to the inhibitory capacities of cinnamaldehyde and 2-methoxy cinnamaldehyde on the canonical NLRP3 inflammasome-mediated IL-1β secretion. However, unexpectedly, at 100 μM, cinnamaldehyde and 2-methoxy cinnamaldehyde significantly enhanced caspase-1 and ASC mRNA expression. We speculated that this may be from a compensatory effect for the shortage of NLRP3.

### 2.3. Influence of Cinnamaldehyde and 2-Methoxy Cinnamaldehyde on Cytosolic Pro-IL-1β, NLRP3 and Pro-Caspase-1 Protein Expression

To confirm the suppressing effect of cinnamaldehyde and 2-methoxy cinnamaldehyde on the maturation stage of IL-1β, the change of protein amount of cytosolic pro-IL-1β, NLRP3 and pro-caspase-1, as well as the secreted mature IL-1β and active caspase-1 were measured and are illustrated in Section 2.3 and Section 2.4. As shown in Figure 5, cytosolic protein expressions of pro-IL-1β and NLRP3 were enhanced after LPS-priming and ATP-activation. At a concentration of 100 μM, cinnamaldehyde and 2-methoxy cinnamaldehyde significantly reduced cytosolic protein amounts of both pro-IL-1β and NLRP3 as compared with those in LPS-primed and ATP-activated THP-1 macrophages. This is consistent with their effect on mRNA expression and pretreatment of cinnamaldehyde. Additionally, 2-methoxy cinnamaldehyde enhanced cytosolic pro-caspase-1 protein levels.

### 2.4. Effect of Cinnamaldehyde and 2-Methoxy Cinnamaldehyde on Secreted IL-1β and Caspase-1 Protein Levels

Due to active caspase-1 and mature IL-1β were secreted immediately to the medium [14], the protein levels of caspase-1 and IL-1β in the conditioned medium were analyzed to observe whether cinnamaldehyde and 2-methoxy cinnamaldehyde affected the activation of caspase-1. As presented in Figure 6, ATP activation markedly elevated IL-1β and active caspase-1 secretion from LPS-primed macrophages. Simultaneously, the level of cytosolic pro-caspase-1 decreased in response to ATP-activation. The secreted IL-1β levels seemed to be parallel with the cytosolic pro-IL-1β levels, while the change of the secreted active caspase-1 level were opposite to the change of the cytosolic procaspase-1 level. Cinnamaldehyde and 2-methoxy cinnamaldehyde at 100 μM diminished ATP-induced IL-1β secretion and caspase-1 release. In contrast, pretreatment of α-methyl cinnamaldehyde did not affect secretion of IL-1β and release of caspase-1. The results revealed that blocking caspase-1 activation seems to be one of the effecting mechanisms of cinnamaldehyde and 2-methoxy cinnamaldehyde to suppress canonical IL-1β secretion.

### 2.5. Direct Inhibitory Effect of Cinnamaldehyde and 2-Methoxy Cinnamaldehyde on Purified Active Caspase-1

To further explore whether cinnamaldehyde and 2-methoxy cinnamaldehyde could directly react with caspase-1 to inhibit enzymatic activity, the purified active caspase-1 was incubated with these tested compounds for 1 h prior to measure the enzymatic activity. As presented in Figure 7, cinnamaldehyde and 2-methoxy cinnamaldehyde inhibited 51.6% and 35.8% catalytic activity of caspase-1 at an extremely high concentration, namely, 1 mM. However, at cellular tolerable concentrations, such as 100 and 50 μM, they did not show any obvious inhibition. The results implied that the inhibition of cinnamaldehyde and 2-methoxy cinnamaldehyde on IL-1β secretion in macrophages was not through the way of directly interfering the caspase-1 catalytic activity.

## 3. Discussion

Our results showed that cinnamaldehyde and 2-methoxy cinnamaldehyde were good at inhibition of canonical NLRP3 inflammasome-mediated IL-1β secretion. However, cinnamic acid, cinnamyl alcohol, cinnamyl acetate and α-methyl cinnamaldehyde almost did not have any inhibitory activity. Apparently, lack of aldehyde group and methyl substitution on α-carbon of the side chain would severely impact the inhibitory capacity. These results implied that the propental group of the side chain was essential for the suppressing ability of cinnamaldehyde-related compounds on the canonical IL-1β secretion. Moreover, 2′-substitution of the benzyl group might fine-tune the inhibitory capacity and consequently, could serve as a target site for synthesis of an effective inhibitor. In fact, the indispensability of propenal group for the anticancer, NO-suppressing, and anti-inflammatory capacities of cinnamaldehyde-related compounds was verified previously by other researchers based on the poor activities of cinnamic acid, cinnamyl alcohol and cinnamyl acetate [9,15,16,17]. Lee et al. indicated that 2′-hydroxycinnamaldehyde isolated from cinnamon had an anti-inflammatory activity more potent than cinnamaldehyde. Furthermore, they further explored the structure-activity relationship by examining the activities of a series of 2′-hydroxycinnamaldehyde derivatives on inhibition of LPS-induced NO generation. Their results concluded that the aldehyde group of the side chain played an essential role in inhibition of NO generation and 2′-substituted groups of the aromatic ring moderately improved the activity [18]. Recently, Ka et al. indicated that two synthetic cinnamaldehyde derivatives, (*E*)-3-phenyl-2-propenoyl-β-d-galactosamine and *N*-cinnamyl-β-d-galactosamine, without a free aldehyde group could not diminish LPS-induced secretion of pro-inflammatory cytokines. In contrast, a 4-glucopyranosyloxy substituted in aromatic ring not only largely reduced cytotoxicity but also retained mostly anti-inflammatory activity of cinnamaldehyde. Furthermore, this compound, 4-hydroxycinnamaldehyde-galatosamine, was documented to ameliorate renal inflammation and IL-1β secretion by attenuating signaling pathways involved in priming and activation of NLRP3 inflammasome [12]. All of the above results indicated that the propenal group played a critical role, while substitution on the aromatic ring played an enhancer in the anti-inflammatory capacity of cinnamaldehyde derivatives.

To date, two structural moieties, the α,β-unsaturated carbonyl and the free aldehyde, have been proposed to be responsible for the biofunctions of cinnamaldehyde analogs, such as anti-inflammatory, anti-cancer and anti-angiogenic activities [9,15,19]. Through Michael addition, the α,β-unsaturated carbonyl moiety can covalently modify cysteine residues in the active site to interfere proteins’ function. The members of inflammatory signaling pathway, including TLR-4, nuclear factor (NF)-κB/p65 and IκB kinase, have been identified as the modified candidates of α,β-unsaturated carbonyl compounds [20,21,22]. In fact, cinnamaldehyde was documented to inhibit LPS-induced TLR-4 dimerization by cysteine modification [10]. Additionally, it was reported that cinnamaldehyde covalently modified cysteine residues of phosphoinositide-3-kinase (PI3K) and phosphoinositide-dependent kinase (PDK)-1, the upstream components of NF-κB signaling, to regulate monocyte/macrophage-mediated immune responses [23]. However, in this study, all tested compounds harbored a α,β-unsaturated keto moiety, while their inhibitory capacities were far different. Macpherson et al. [24] indicated that a carbonyl substitution adjacent to the enone would booster a more reactive cinnamaldehyde-like Michael acceptor to activate transient receptor potential ankyrin 1 (TRPA1). In contrast, some structural analogues, such as propionaldehyde and cinnamic alcohol, are chemically inert Michael acceptors and cannot activate TRPA1. Based on the above ideas, we propose that aldehyde group loss and/or α-methyl substitution of the side chain would severely diminish the reactivity of Michael addition of cinnamaldehyde-related compounds, and thereby, influence their bioactivities. Correspondingly, currently developed theories of small molecular inflammasome inhibitors refer almost identically to the concepts that encompass Michael acceptors with potent reactivity [25]. Moreover, peptide aldehydes were indicated to be able to react with catalytical hydroxyl or thiol groups in the active site of serine and cysteine proteases to form a reversible hemi(thio)acetal, and thereby, to inhibit catalytic activity of these proteases [26]. Based on the poor inhibitory capacities on caspase-1 catalytic activity, it is impossible for the aldehyde group of cinnamaldehyde and 2-methoxy cinnamaldehyde to react with the catalytic hydroxyl or thiol groups of caspase-1 to influence IL-1β maturation. However, it could not exclude the possibility that the aldehyde group of cinnamaldehyde react with hydroxyl or thiol groups of other critical proteins involved in NLRP3 inflammasome-mediated IL-1β secretion.

Additionally, our results showed that cinnamaldehyde and 2-methoxy cinnamaldehyde attenuated mRNA and protein expressions of NLRP3 and IL-1β at the priming stage. Simultaneously, cinnamaldehyde and 2-methoxy cinnamaldehyde retarded ATP-induced conversion of procaspase-1 into active caspase-1. However, a direct inhibition on caspase-1 catalytic activity seemed to be impossible because of the extraordinary concentrations required. Undoubtedly, the inhibitory effect of cinnamaldehyde and 2-methoxy cinnamaldehyde on the priming stage of IL-1β secretion is attributed to their well-known NF-κB suppressing capacities [18]. Recently, cinnamaldehyde was demonstrated by some researchers to attenuate IL-1β secretion by attenuating NLRP3 inflammasome activation, and subsequently, alleviate inappropriate inflammatory response in heart of fructose-induced metabolic rats as well as in lung and kidney of endotoxin-poisoned mice [27,28,29]. Through amelioration of cardiac oxidative stress, cinnamaldehyde retarded CD36-mediated TLR4/6-IRAK4/1 signaling and then to stifle NLRP3 inflammasome activation in fructose-caused myocardial cell inflammation [27]. Cinnamaldehyde was proposed by some investigators to inhibit NLRP3 inflammasome activation through three intracellular events: attenuating P2X7R expression and K^+^ efflux, reducing ROS production, as well as decreasing lysosomal rupture and capthesin B release [28]. Although a detailed description of this mechanism is needed to explore it further, our results showed that cinnamaldehyde and 2-methoxy cinnamaldehyde exerted a potent dual suppressing effect on canonical IL-1β secretion at both stages of priming and activation.

On the basis of our current results, we propose that the propenal group of cinnamaldehyde-related compounds plays a decisive role in inhibition of canonical IL-1β secretion and substituted group in the aromatic ring may serve as an adjuster during the inhibition process. Cinnamaldehyde and 2-methoxy cinnamaldehyde inhibited canonical NLRP3 inflammasome-mediated IL-1β secretion through suppressing both steps of priming and activation. Thus, these cinnamaldehyde-related compounds would have an excellent potential to serve as dietary supplements for the treatment of sterile inflammation-related diseases.

## 4. Materials and Methods 

### 4.1. Chemicals and Reagents

Cinnamon phytochemicals, including cinnamic acid, cinnamyl alcohol, cinnamyl acetate, cinnamaldehyde, 2-methoxy cinnamaldehyde and α-methyl cinnamaldehyde, dimethyl sulfoxide (DMSO), phorbol 12-myristate 13-acetate (PMA), LPS (Escherichia coli O55:B5), adenosine triphosphate (ATP), 3-(4,5-dimethyldiazol-2-yl)-2,5-diphenyl tetrazolium bromide (MTT) and trichloroacetic acid (TCA) were procured from Sigma-Aldrich (St. Louis, MO, USA). The medium and reagents for cell culture were purchased from Thermo Fisher Scientific (Grand Island, NE, USA). Chemicals of analytical grade were used.

### 4.2. Culture and Differentiation

THP-1 human leukemic monocytes acquired from the Bioresource Collection and Research Center (Hsinchu, Taiwan) were grown in a 1640 medium added with 10% fetal bovine serum, 100 μg/mL streptomycin, and 100 U/mL penicillin. THP-1 monocytes were differentiated into macrophages by a treatment with 100 nM PMA for 24 h, and then fasted overnight by incubation with fresh medium. The cytotoxic effect of the tested cinnamaldehyde related compounds on THP-1 macrophages were evaluated by using an MTT method. Following a 24 h treatment of the cinnamaldehyde related compounds at concentrations from 25 to 200 μM, the THP-1 macrophages were incubated with MTT (0.5 mg/mL) for 4 h. After being dissolved in DMSO, the generated formazan was measured with an absorbance at 550 nm. The viability of cells incubated with tested compound was expressed as the percentage (%) of cells treated with a DMSO vehicle. The tested compounds were dissolved in DMSO to prepare stock solutions of 100 mM. Consequently, the highest final concentration of DMSO vehicle in culture medium did not exceed 0.2%, and did not influence cell viability.

### 4.3. Stimulation and Analysis of IL-1β and TNF-α

To explore the suppressing effect of cinnamaldehyde-related compounds on entire two-stage canonical IL-1β secretions, the macrophages were subjected, in sequence, with tested compounds (25~100 μM) for 30 min, 1 μg/mL LPS for 4 h and 5 mM ATP for 3 h. To evaluate the suppressing effect of these compounds on the second stage, namely NLRP3 inflammasome activation, the macrophages were primed with LPS for 4 h. After a PBS wash, the LPS-primed cells were then incubated with tested compounds (25~100 μM) for 30 min and ATP for 3 h. The condition medium was pick up and stored at −20 °C for further assay. An enzyme-linked immunosorbent assay (ELISA) (BioLegend, San Diego, CA, USA) was applied to measure medium concentrations of IL-1β and TNF-α.

### 4.4. RNA Isolation and Quantitative RT-PCR

After the treatment with the indicated compounds and LPS priming, the cells were harvested for RNA isolation. The cellular RNA was first isolated with Trizol reagent (Invitrogen, Carlsbad, CA, USA), and then reverse-transcribed to cDNA by using a commercial kit (Promega, Madison, WI, USA). The quantitative PCR of cDNA was performed on a StepOne^TM^ real-time PCR system (Applied Biosystems, Foster City, CA, USA). The sequences of primer pairs used in quantitative PCR were described in our previous study [30]. The 2^−ΔΔCT^ method was applied to calculate the relative mRNA expression of target gene to the reference gene (β-actin) [31]. 

### 4.5. Secreted and Cytosolic Protein Preparation, and Western Blotting

The macrophages were sequentially incubated with the indicated compounds, LPS and ATP. The proteins in the collected medium were precipitated by adding TCA to an ultimate concentration of 5%. After being centrifuged at 12,000× *g* for 10 min, the pellets were dissolved in sample buffer (60 mM Tris-Cl pH 6.8, 2% SDS, 5% β-mercaptoethanol, 10% glycerol, 0.01% bromophenol blue) and boiled for 10 min. A commercial kit (Thermo Fisher Scientific) was adopted to isolate the cytosolic proteins. Expressions of target proteins were then detected by immunoblotting. In brief, samples with equal amount of proteins were subjected into a SDS-PAGE gel and eletrophorezised. The separated protein bands were transblotted onto a polyvinylidene difluoride (PVDF) filter, and then detected with antibodies against NLRP3 (Novus Biological, Littleton, CO, USA), caspase-1 (Santa Cruz Biotechnology, Dallas, TX, USA), IL-1β (Abcam, Cambridge, MA, USA) and β-actin (Biovision, Milpitas, CA, USA). Finally, the chemiluminescence of immunoreactive protein bands were captured with a chemiluminescence imager. The Image J software (National Institutes of Health, Bethesda, MD, USA) was adopted to quantify the densitometry of each target protein band and the expression of target protein was normalized with β-actin.

### 4.6. Measurement of Inhibitory Capacities on the Caspase-1 Catalytic Activity

The inhibitory capacities of the indicated compounds on the enzymatic activity of caspase-1 was detected with a commercial kit (Abcam). The assay uses a specific peptide substrate, YVAD-AFC (7-amino-4-trifluoromethyl coumarin), which is cleaved by caspase-1 to release free fluorescent AFC. The active caspase-1 were incubated with cinnamon phytochemicals and YVAD-AFC substrate in the reaction buffer at 37 °C for 1 h. Then, the fluorescence intensity of fluorescent AFC was measured at a setting of 400 nm excitation wavelength and 505 nm emission wavelength by using a multimode reader. The inhibitory capacity of tested compound was expressed as the percentage (%) of reduction on caspase-1 activity.

### 4.7. Statistical Analysis

All data were expressed as mean ± SD. In the cytotoxic experiment, student t-test was used to analyze the difference between treatments and control. In other experiments, statistical comparisons among different treatments at the top testing concentration were analyzed by a one way ANOVA followed by Duncan’s test. The statistical significance was set at *p* < 0.05. Statistical analyses were carried out with the SPSS 22.0 software (IBM, Armonk, NY, USA).

## Figures and Tables

**Figure 1 molecules-23-03241-f001:**
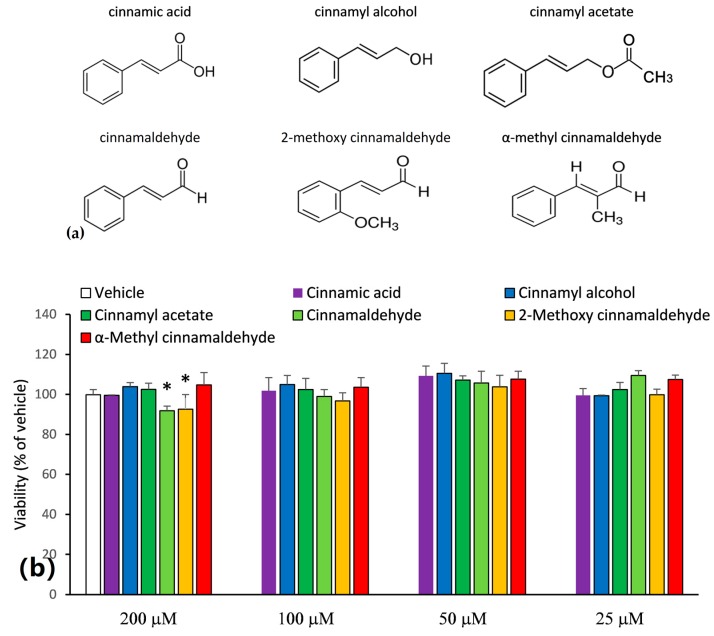
(**a**) Chemical structures of the cinnamaldehyde-related compounds; (**b**) Cytotoxic effect of cinnamaldehyde-related compounds on THP-1 macrophages. The macrophages were incubated with cinnamaldehyde-related compounds at 25~200 μM for 24 h. The viabilities were then evaluated by MTT method and expressed as % of dimethyl sulfoxide (DMSO) vehicle control. * represents statistically significant differences (*p* < 0.05) between treatment group and vehicle control.

**Figure 2 molecules-23-03241-f002:**
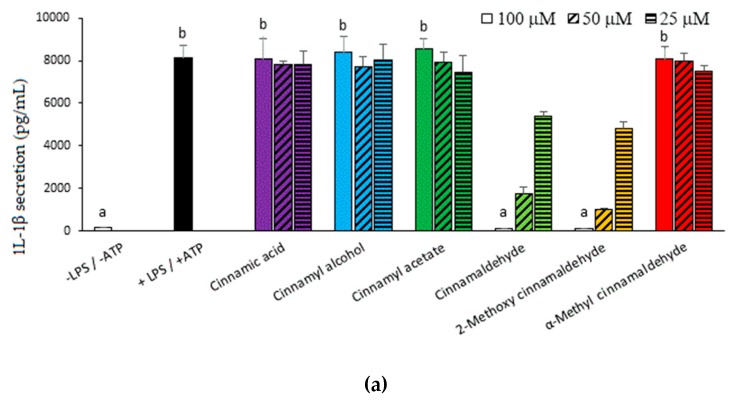
Influence of cinnamaldehyde-related compounds on canonical interleukin (IL)-1β (**a**) and TNF-α secretion (**b**). The macrophages were pretreated with 100 μM cinnamaldehyde-related compounds for 30 min and then treated with lipopolysaccharide (LPS) and adenosine 5′-triphosphate (ATP). Data are expressed as the mean ± SD and different superscript letters among 100 μM treatment groups represent statistically significant differences (*p* < 0.05).

**Figure 3 molecules-23-03241-f003:**
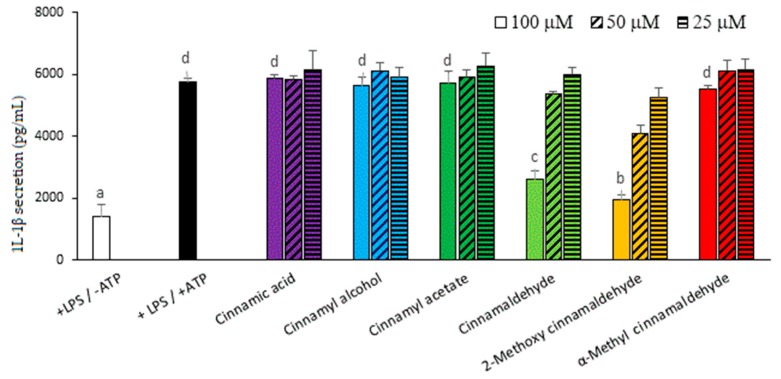
Influence of cinnamaldehyde-related compounds on ATP-induced IL-1β secretion in LPS-primed cells. The LPS-primed macrophages were treated with 100 μM cinnamaldehyde-related compounds for 30 min and then activated with ATP. Data are expressed as the mean ± SD and different superscript letters among 100 μM treatment groups represent statistically differences (*p* < 0.05).

**Figure 4 molecules-23-03241-f004:**
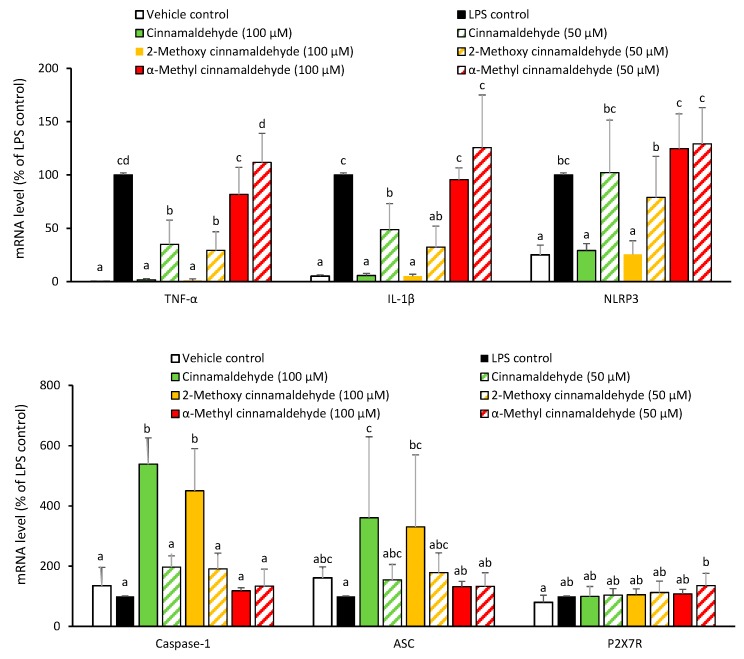
Influence of cinnamaldehyde-related compounds on LPS-primed TNF-α, IL-1β, NOD-like receptor protein 3 (NLRP3), caspase-1, adaptor proteins- apoptosis-associated speck-like containing CARD (ASC) and P2X7 receptor (P2X7R) mRNA expressions. The macrophages were pretreated with 100 μM cinnamaldehyde-related compounds for 30 min and then treated with LPS. The mRNA levels were detected by RT-qPCR. Data are expressed as the mean ± SD and different superscript letters represent statistically differences (*p* < 0.05).

**Figure 5 molecules-23-03241-f005:**
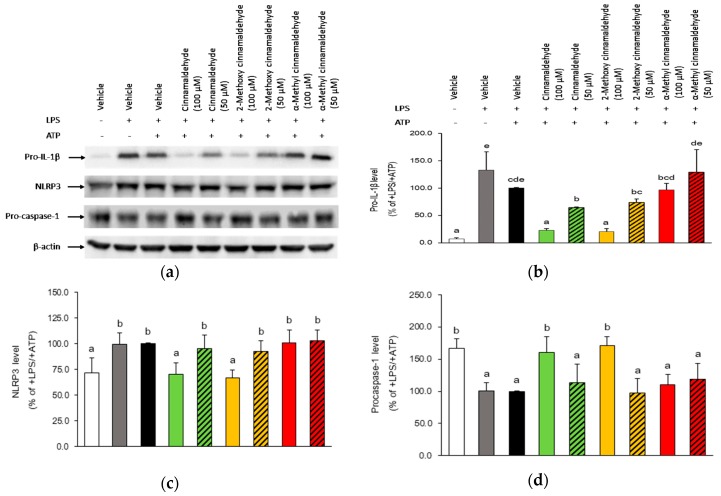
Influence of cinnamaldehyde-related compounds on the cytosolic pro-IL-1β, NLRP3 and pro-caspase-1 levels in macrophages stimulated with LPS and ATP. The panels display a representative photogram of Western blot (**a**) and the quantified results of pro-IL-1β (**b**), NLRP3 (**c**) and pro-caspase-1 (**d**), separately. The macrophages were pretreated with 100 μM cinnamaldehyde-related compounds for 30 min and then treated with LPS and ATP. Cytosolic proteins were isolated and the pro-IL-1β, NLRP3 and pro-caspase-1 amounts were further detected by Western blot. Data are expressed as the mean ± SD and different superscript letters represent statistically differences (*p* < 0.05).

**Figure 6 molecules-23-03241-f006:**
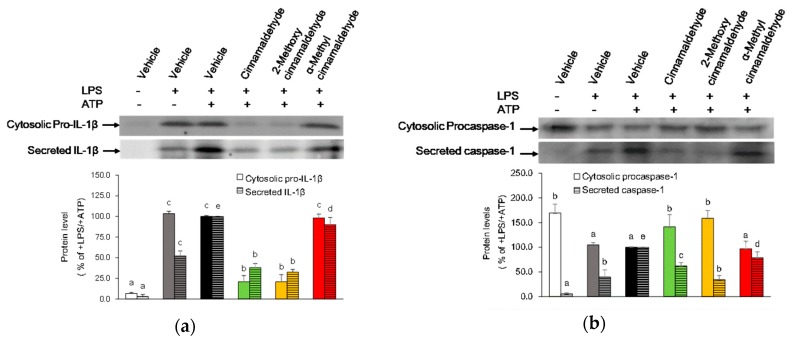
Influence of cinnamaldehyde-related compounds on the secreted protein levels of IL-1β (**a**) and caspase-1 (**b**) from LPS-primed and ATP-activated macrophages. The macrophages were pretreated with 100 μM cinnamaldehyde-related compounds for 30 min and then treated with LPS and ATP. Medium proteins were collected to measure protein levels of IL-1β and caspase-1 by Western blot. Data are expressed as the mean ± SD and different superscript letters presented statistical difference (*p* < 0.05).

**Figure 7 molecules-23-03241-f007:**
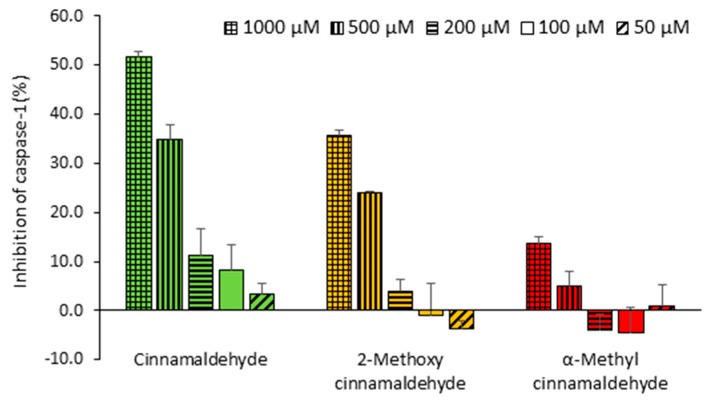
Influence of cinnamaldehyde-related compounds on the caspase-1 catalytic activity. The active caspase-1 were incubated with cinnamaldehyde-related compounds at concentrations from 50 to 1000 μM for 1 h. The suppressing ability was expressed as % of reduction on caspase-1 catalytic activity.

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
