# Peer review of "Structural Moieties Required for Cinnamaldehyde-Related Compounds to Inhibit Canonical IL-1β Secretion"

_molecules, 2018, doi:10.3390/molecules23123241_

Round 1
Reviewer 1 Report
The manuscript by Ho et al. reported biological evaluations of six cinnamaldehyde-related compounds on the inhibition of NLRP3 inflammasome and related responses. They found that cinnamaldehyde and 2-methoxy cinnamaldehyde are most active that can suppress IL-1β, TNF-α, NLRP3, and caspase-1 at mRNA and protein levels in macrophages. They also explore the potential SAR responsible for the biological effects. The manuscript is well organized and written, and it is a good addition to the cinnamaldehyde biochemistry. Some comments would help with the improvement of the manuscript.
1) The authors should consider re-draw the bar charts in the figures, with better color codes and legends. Figures 2, 3, 6 are hard to read.
2) It would be interesting to see how new cinnamaldehyde derivatives improve the bioactivities based on the current SAR.
Author Response
Reviewer #1:
Comments: The manuscript by Ho et al. reported biological evaluations of six cinnamaldehyde-related compounds on the inhibition of NLRP3 inflammasome and related responses. They found that cinnamaldehyde and 2-methoxy cinnamaldehyde are most active that can suppress IL-1β, TNF-α, NLRP3, and caspase-1 at mRNA and protein levels in macrophages. They also explore the potential SAR responsible for the biological effects. The manuscript is well organized and written, and it is a good addition to the cinnamaldehyde biochemistry. Some comments would help with the improvement of the manuscript.
1) The authors should consider re-draw the bar charts in the figures, with better color codes and legends. Figures 2, 3, 6 are hard to read.
Response:
Thanks for your suggestion. We replaced the bar pattern and hope that the figures will be more clear to readers.
2) It would be interesting to see how new cinnamaldehyde derivatives improve the bioactivities based on the current SAR.
Response:
Thanks for your comments. We also expect that our results will aid the development of new cinnamaldehyde derivatives with more strong bioactivities.
Reviewer 2 Report
The manuscript by Ho et al. evaluated six cinnamaldehyde-related compounds
The manuscript by Ho et al. evaluated six cinnamaldehyde-related compounds on canonical inflammasome activation in an attempt to establish structural requirements needed for inhibition in THP-1 macrophages. Only two compounds namely cinnamaldehyde and 2-methoxy cinanamaldehide exerted inhibitory activity on canonical NLRP3 inflammasome-mediated Il-1β secretion.
The anti-inflammatory activity of cinnamaladehyde through inhibition of NF-kB has been previously reported in the literature. The manuscript needs major revision and some questions should be addressed by the authors.
The authors should revise numeration of figures. There are two Figures 1.
1. The compounds are tested at very high concentrations (25-200 µM). Are these compounds used as food additives at high concentrations?.
2. Cytotoxicity of these compounds is very relevant. In Fig.1 “cytotoxic effects of cinnamaladehyde-related compounds”, viability is expressed as % of vehicle (DMSO). DMSO is toxic to cells. Which concentration of DMSO was used?. The authors do not mention how these compounds were dissolved. Cell viability values for DMSO-treated cells should be referred.
3. Fig.2 Effects on LPS-primed and ATP-activated IL-1B. The authors stated that cells were primed with LPS for 4 h, but later they incubated for 3 h instead of 4 h (see line 124).
4. Fig. 3 Statistics are very confusing. The authors should revised the statistical analysis.
5. Line 149: “In consistent with” should be replaced by “This is consistent with”. English re-edition of the manuscript is recommended.
6. Fig 5a. Western-blot of secreted IL-1β. Stimulation of cells with LPS + vehicle shows a band, which means that IL-1β is already secreted when cells are stimulated with LPS. But, it is also secreted in the presence of LPS+ ATP. An explanation for this result should be provided.
7. In all experiments performed, macrophages were incubated with compounds for 30 min, primed with LPS for 4 h. and then activated with ATP for 3 h. Additional experiments should be performed with compounds incubated after LPS stimulation and before adding ATP, to confirm that compounds exert inhibitory effects at the second stage (ATP-activation). Authors concluded (see lines 264-267) that the two active compounds exerted dual inhibitory effects at both stages of priming and activation, but this should be demonstrated.
Author Response
Reviewer #2
The manuscript by Ho et al. evaluated six cinnamaldehyde-related compounds on canonical inflammasome activation in an attempt to establish structural requirements needed for inhibition in THP-1 macrophages. Only two compounds namely cinnamaldehyde and 2-methoxy cinanamaldehide exerted inhibitory activity on canonical NLRP3 inflammasome-mediated IL-1β secretion.
The anti-inflammatory activity of cinnamaladehyde through inhibition of NF-kB has been previously reported in the literature. The manuscript needs major revision and some questions should be addressed by the authors.
The authors should revise numeration of figures. There are two Figures 1.
Response:
Thanks for your careful review. We have revised the number of the second figure.
1. The compounds are tested at very high concentrations (25-200 µM). Are these compounds used as food additives at high concentrations?
Response:
In fact, the testing highest concentration of these compounds was 100 µM (13.2 mg/L).
Among the test compounds, cinnamaldehyde is the principal aroma and flavour component of cinnamon. Cinnamon is a common spice and widely used in various cuisines, sweet and savoury dishes, breakfast cereals, snack foods, tea and traditional foods. There are no recommendation and limitation on the use of cinnamon in foods, and thereby, we can’t provide a clear-cut used concentration of cinnamaldehyde. However, we can provide some information regarding its use in animal study. For example, Xu et al., (2017) fed rats daily a single oral dose (0.132 or 0.264 g/kg BW) of cinnamaldehyde. In study of Lee et al. (2017), rats received a dose of 0.45 or 0.9 mg/kg BW of cinnamaldehyde by gavage. In study of Liao et al. (2012), the cinnamaldehyde dose of intraperitoneal injection were 1.25-5 mg/kg.
Although the testing concentration is high, it did not affect cell viability. And we think the results herein are reasonable, solid and enough to support the conclusion.
2. Cytotoxicity of these compounds is very relevant. In Fig.1 “cytotoxic effects of cinnamaladehyde-related compounds”, viability is expressed as % of vehicle (DMSO). DMSO is toxic to cells. Which concentration of DMSO was used? The authors do not mention how these compounds were dissolved. Cell viability values for DMSO-treated cells should be referred.
Response:
The tested compounds were dissolved in DMSO to prepare stock solutions of 100 mM. The tested compounds were dissolved in DMSO to prepare stock solutions of 100 mM. Consequently, the highest final concentration of DMSO vehicle in culture medium did not exceed 0.2 % (0.2% in the cytotoxic experiment and 0.1% in the others), and did not influence cell viability.
3. Fig.2 Effects on LPS-primed and ATP-activated IL-1B. The authors stated that cells were primed with LPS for 4 h, but later they incubated for 3 h instead of 4 h (see line 124).
Response:
Thanks for your careful review. We have revised the typing error.
4. Fig. 3 Statistics are very confusing. The authors should revised the statistical analysis.
Response:
Thanks for your comments. We revised section 4.7 as following and added the statistical explanation in the figure legends to avoid confusing.
4.7. Statistical analysis
All data were expressed as mean ± S.D. In cytotoxic experiment, student t-test was used to analyze the difference between treatments and control. In other experiments, statistical comparisons among different treatments at the top testing concentration were analyzed by the one way ANOVA followed by Duncan's test. The statistical significance was set at p<0.05. Statistical analyses were carried out with the SPSS 22.0 software.
5. Line 149: “In consistent with” should be replaced by “This is consistent with”. English re-edition of the manuscript is recommended.
Response:
Thanks for your comments. We revised the sentence which you mentioned. Additionally, the manuscript was thoroughly be re-edit.
6. Fig 5a. Western-blot of secreted IL-1β. Stimulation of cells with LPS + vehicle shows a band, which means that IL-1β is already secreted when cells are stimulated with LPS. But, it is also secreted in the presence of LPS+ ATP. An explanation for this result should be provided.
Response:
Thanks for your comments. In addition to canonical pathway, a Fas-dependent non-canonical IL-1β secretion is documented in LPS-treated cells. LPS can trigger activation of Fas signaling which further activate caspase-8 through the adapter molecule Fas-associated death domain (FADD). The active capase-8 cleavage immature pro-IL-1 into mature IL-1. Therefore, we added following explanation in the revised manuscript.
In fact, in addition to the canonical pathway, a non-canonical IL-1β secretion has been demonstrated [13]. The TRL ligands, such as LPS, can trigger activation of Fas signaling which will further activate caspase-8 through the adapter molecule Fas-associated death domain (FADD). The active capase-8 then cleavage immature pro-IL-1β into mature IL-1β. Consequently, as shown here, a single LPS-priming was enough to make macrophages to secret IL-1β through non-canonical pathway.
13. Bossaller, L.; Chiang, P.I.; Schmidt-Lauber, C.; Ganesan, S.; Kaiser, W.J.; Rathinam, V.A.K.; Mocarski, E.S.; Subramanian, D.; Green, D.R.; Silverman, N.; Fitzgerald, K.A. Marshak-Rothstein, A.; Latz, E. FAS mediates non-canonical IL-1β and IL-18 maturation via caspase-8 in a Rip3-independent manner. J. Immunol. 2012, 189(12), 5508–5512.
7. In all experiments performed, macrophages were incubated with compounds for 30 min, primed with LPS for 4 h. and then activated with ATP for 3 h. Additional experiments should be performed with compounds incubated after LPS stimulation and before adding ATP, to confirm that compounds exert inhibitory effects at the second stage (ATP-activation). Authors concluded (see lines 264-267) that the two active compounds exerted dual inhibitory effects at both stages of priming and activation, but this should be demonstrated.
Response:
Thanks for your comments. We added the results in section 2.1 as follow.
In order to explore the inhibitory effect of these compounds on the maturation stage of canonical pathway of IL-1β, macrophages were primed firstly before compound treatment and ATP activation. As shown in Figure 3, without ATP activation, LPS-primed cells could secret a moderate amount (1415 pg/mL) of IL-1β. In fact, in addition to the canonical pathway, a non-canonical IL-1β secretion has been demonstrated [13]. The TRL ligands, such as LPS, can trigger activation of Fas signaling which will further activate caspase-8 through the adapter molecule Fas-associated death domain (FADD). The active capase-8 then cleavage immature pro-IL-1β into mature IL-1β. Consequently, as shown here, a single LPS-priming was enough to make macrophages to secret IL-1β through non-canonical pathway. Despite of this, ATP activation further enhanced IL-1β secretion to 5745 pg/mL and this increased IL-β should originate from canonical NLRP3 inflammasome pathway. Among the tested compounds, only cinnamaldehyde and 2-methoxy cinnamaldehyde significantly attenuated ATP-stimulated IL-1β secretion. The results implied that cinnamaldehyde and 2-methoxy cinnamaldehyde could inhibit partially maturation process of canonical pathway.
Round 2
Reviewer 2 Report
No new questions are required